# Computational analysis of long-range allosteric communications in CFTR

**Ayca Ersoy[1,2†], Bengi Altintel[1,2†], Nurit Livnat Levanon[3†], Nir Ben-Tal[4], Turkan Haliloglu[1,2]\*, Oded Lewinson[3]\***

[1]Department of Chemical Engineering, Bogazici University, Istanbul, Turkey; [2]Polymer Research Center, Bogazici University, Istanbul, Turkey; [3]Department of Molecular Microbiology, Bruce and Ruth Rappaport Faculty of Medicine, Technion-Israel Institute of Technology, Tel Aviv, Israel; [4]Department of Biochemistry and Molecular Biology, Faculty of Life Sciences, Tel-Aviv University, Tel-Aviv, Israel

**\*For correspondence:**
halilogt@boun.edu.tr (TH);
lewinson@technion.ac.il (OL)

[†]These authors contributed equally to this work

**Competing interest:** The authors declare that no competing interests exist.

**Abstract** Malfunction of the CFTR protein results in cystic fibrosis, one of the most common hereditary diseases. CFTR functions as an anion channel, the gating of which is controlled by long-range allosteric communications. Allostery also has direct bearings on CF treatment: the most effective CFTR drugs modulate its activity allosterically. Herein, we integrated Gaussian network model, transfer entropy, and anisotropic normal mode-Langevin dynamics and investigated the allosteric communications network of CFTR. The results are in remarkable agreement with experimental observations and mutational analysis and provide extensive novel insight. We identified residues that serve as pivotal allosteric sources and transducers, many of which correspond to disease-causing mutations. We find that in the ATP-free form, dynamic fluctuations of the residues that comprise the ATP-binding sites facilitate the initial binding of the nucleotide. Subsequent binding of ATP then brings to the fore and focuses on dynamic fluctuations that were present in a latent and diffuse form in the absence of ATP. We demonstrate that drugs that potentiate CFTR's conductance do so not by directly acting on the gating residues, but rather by mimicking the allosteric signal sent by the ATP-binding sites. We have also uncovered a previously undiscovered allosteric 'hotspot' located proximal to the docking site of the phosphorylated regulatory (R) domain, thereby establishing a molecular foundation for its phosphorylation-dependent excitatory role. This study unveils the molecular underpinnings of allosteric connectivity within CFTR and highlights a novel allosteric 'hotspot' that could serve as a promising target for the development of novel therapeutic interventions.

## eLife assessment

This manuscript presents a **useful** analysis of allosteric communication in the CFTR protein using a coarse-grained dynamic model and characterized the role of disease-causing mutations. The results and analyses are generally **solid** and validated with available experimental observations. The findings provide comprehensive insights into the allosteric mechanism of this protein.

## Introduction

Cystic fibrosis (CF) is a multifaceted disorder impacting the epithelial tissues of various organs, including the lungs, intestinal tract, and pancreas. It stands as one of the prevailing hereditary afflictions, affecting approximately 1 in every 3000 newborns (*Farrell, 2008*). Notwithstanding the significant advancements in therapeutic interventions, individuals with CF continue to endure pronounced morbidity and a reduced life expectancy (*McBennett et al., 2022*). The disease is caused by mutations

in the gene encoding the cystic fibrosis transmembrane conductance regulator (CFTR) protein. Most CF patients carry a deletion of Phe 508 in nucleotide binding domain (NBD)1 which leads to protein instability and premature degradation. More than 2000 other disease-associated mutations have been documented (*Cftr2.org, 2011*), many of which remain uncharacterized. CFTR functions as a chloride channel, the excretion of which is important for mucus hydration. Malfunction of CFTR leads to mucus dehydration and a subsequent decrease in mucus clearance. The over-accumulation of mucus leads to persistent bacterial infections, chronic inflammations, and to organ failure (*Fahy and Dickey, 2010*). CFTR is a member of the ABC transporter protein family, considered to be one of the largest protein families of any proteome (*Dean et al., 2001*). CFTR is a unique family member, as all other ABC transporters function as active pumps, moving biomolecules against their concentration gradients (*Rees et al., 2009*; *Gadsby et al., 2006*). The domain organization of CFTR is similar to that of canonical ABC transporters, comprising two transmembrane domains (TMDs) and two cytosolic NBDs. The TMDs form the translocation channel and the NBDs control channel opening and closing through processive cycles of ATP binding, hydrolysis, and release. A unique feature of CFTR is the regulatory domain (R) which contains multiple phosphorylation sites and connects the two NBDs giving rise to a TMD1-NBD1-R-TMD2-NBD2 topology. Decades of structure-function studies enabled the formalization of a mechanistic model for the function of CFTR: In the resting (non-conducting) state, the funnel-shaped pore is inward-facing and is solvent accessible only to the cytoplasm. The NBDs are separated, and the R domain is wedged between them, preventing their dimerization (*Liu et al., 2017b*; *Zhang and Chen, 2016*). Upon phosphorylation of the R domain by PKA, it disengages from its inhibitory position and shifts aside (*Picciotto et al., 1992*; *Tabcharani et al., 1991*). The NBDs are now free to bind ATP and form a "head-to-tail" dimer (*Rees et al., 2009*; *Zhang et al., 2018*; *Mense et al., 2006*), with the two ATP molecules sandwiched at composite binding sites formed at the dimer interface. In CFTR, like in some other ABC transporters, only one of the ATP sites is hydrolysis-competent while the other (termed the degenerate site) can only bind ATP but not hydrolyze it (*Hohl et al., 2012*; *Csanády et al., 2019*; *Stockner et al., 2020*). Formation of the NBDs dimer leads to rearrangements of transmembrane gating residues and subsequent channel opening. Hydrolysis of ATP at the catalytic site and release of hydrolysis products leads to opening of the NBDs, channel closure, and resetting of the system. This mechanism bears resemblance to that of bona fide ABC transporters, where ATP binding and hydrolysis at the NBDs drive the conversion of the TMDs between inward- and outward-facing conformations (*Johnson and Chen, 2018*; *Kim and Chen, 2018*; *Nguyen et al., 2018*).

It is now broadly recognized that most proteins and enzymes are allosteric, where ligand binding at one site affects the functional properties of another (often distant) site (*Kessel and Ben-Tal, 2018*; *Liu and Nussinov, 2016*; *Gerek and Ozkan, 2011*; *Campitelli et al., 2020*). Two conceptual models dominated the field of allostery for the past 50 years. According to Koshland, Némethy, and Filmer (KNF model, *Koshland et al., 1966*), the protein exists in a single conformation (an inactive one) to which the allosteric ligand binds, inducing a conformational change that leads to the active conformation. In contrast, Monod, Wyman, and Changeux (MWC model, *Monod et al., 1965*) introduced the concept of 'conformational selection', according to which proteins pre-exist in multiple conformation, and the allosteric ligand preferentially binds to the active conformer, leading to a redistribution of the conformational distributions toward the active one (*Ma et al., 1999*). The simplicity of these models is appealing, yet we now begin to understand that they alone cannot explain the complexity of allosteric signal generation and transduction. More recently, an alternative allosteric mechanism was demonstrated, termed 'dynamic mechanism' (*Motlagh et al., 2014*; *Nussinov and Tsai, 2015*), where allostery is mediated by changes in dynamic fluctuations around a mean conformation in the absence of a noticeable conformational change (*Popovych et al., 2006*; *Tsai et al., 2008*). In some proteins, as we believe is the case of CFTR, multiple allosteric mechanisms combine to generate the observed allosteric phenomenon.

There are abundant examples that long-range allosteric communications are essential to the function of CFTR. One such example is the control of channel opening and closing, where the binding site of the effector molecule (ATP) and the affected gating residues are separated by 60–70 Å (*Liu et al., 2017b*; *Zhang et al., 2018*). Very recently, Levring et al. demonstrated that ATP-dependent dimerization of the NBDs is insufficient for channel opening which additionally depends on an allosteric pathway that connects the NBDs and the channel pore (*Levring et al., 2023*). Based on rate-equilibrium free-energy relationship analysis, Sorum et al. reached similar conclusions, and suggested

that this allosteric signal originates at the NBDs and gradually propagates toward the gating residues along a clear longitudinal protein axis (**Sorum et al., 2015**). Another example of long-range allostery in CFTR is the finding that point mutations at the cytosolic loops increase channel open probability (even in the absence of ATP) by shifting a pre-existing conformational equilibrium toward the ligand bound conducting conformation (**Wang et al., 2010**). Notably, allostery in CFTR has direct therapeutic relevance: The potentiator drug ivacaftor and the dual-function modulator elexacaftor do not bind in the proximity of the gating residues (**Liu et al., 2019**; **Fiedorczuk and Chen, 2022a**) yet increase the open probability (Po) of the channel (**Eckford et al., 2012**; **Shaughnessy et al., 2021**). These and other studies (**Sorum et al., 2017**; **Scholl et al., 2021**; **Wei et al., 2014**) significantly advanced our understanding of the function of CFTR and its allosteric control.

However, some important features of the underlying allosteric network remain unknown. For example, the complete allosteric map of CFTR remains to be determined, as are the causal relations (driving vs. driven) between residues and domains. Similarly, the allosteric mechanism for modulation of CFTR by commercial drugs is only partially understood. Herein, we sought to further our understanding of the allosteric network of CFTR. For this, we combined two complementary computational approaches: Gaussian network model-transfer entropy (GNM-TE; **Hacisuleyman and Erman, 2017a**; **Altintel et al., 2022**) and ANM-LD simulations which combine anisotropic network model (ANM) (**Atilgan et al., 2001**; **Wang et al., 2004**; **Keskin et al., 2002**; **Keskin, 2007**) and Langevin dynamics (LD) (**Brünger et al., 1984**).

GNM reduces the complexity of a protein polymer to a network of nodes (representing the main chain α-carbons) connected by springs with a spring constant of unity. Further simplifications are the assumptions of force linearity and isotropy of all fluctuations. These simplifications are the source of both its advantages and limitations. On the one hand, GNM is computationally economical, allowing for rapid analytical derivation of conformational dynamics even of large proteins. This computational economy, while offering valuable insights, does entail a limitation. GNM proves highly effective in predicting fluctuations around a predefined native state and offers valuable information regarding allosteric communication and causality. Nevertheless, due to its reliance on topology, GNM is unsuitable to monitor the conformational transitions from one native state to another. However, it does have the capacity to suggest potential modes of motion accessible within a given protein topology that may contribute to such conformational changes. In addition, GNM cannot provide any information on side chains dynamics as these are omitted from the model. The robustness of the model makes GNM a highly effective tool in studying global dynamics. Despite, or because of, its simplicity, GNM-based approaches have been very successful in describing global conformational dynamics of proteins (**Yang and Bahar, 2005**; **Li et al., 2014**; **Acar et al., 2020**). More details on the fundamentals of GNM, its advantages and limitations can be found here (**Cui and Ivet, 2006**). To supplement GNM we integrated it with TE calculations. The concept of TE is derived from information theory and is used to measure the amount of information transferred between two processes (**Hacisuleyman and Erman, 2017a**). In the context of proteins' conformational changes, TE can be used to determine whether two residues are allosterically connected (**Altintel et al., 2022**). TE for each pair of i and j residues can be viewed as the degree to which the present movement of residue i decreases the uncertainty regarding the future movement of residue j within a specified time delay $\tau$. If TEij($\tau$)>TEji($\tau$), then the dynamics of residue i affects the dynamics of residue j, representing a causal relationship between residues i and j. If a residue transfers information to many other residues in the protein, it is considered as an information source; if it accepts information from much of the protein, it is considered as an information receiver or a 'sink' (for more details about the correlation between TE and allostery, see **Hacisuleyman and Erman, 2017b**). In the present work, we performed TE calculations based on GNM, which considers harmonic interactions between residue pairs (**Hacisuleyman and Erman, 2017a**; **Acar et al., 2020**). Unlike simulation trajectory approaches that infer cause-and-effect relations from the observed sequence of events (which may be circumstantial), GNM-based TE provides a direct measure, at the linear approximation, of causality relations between residues and domains. Quantitative inference of cause-and-effect relations from simulation trajectory approaches is in principle possible using enhanced kinetic sampling techniques but is highly computationally demanding for a large transmembrane transporter such as CFTR. Here, we opted to use a complimentary GNM-based TE approach that readily provides a quantitative measure of sources and receivers of allosteric signals. The code for GNM-TE is freely available on GitHub, (copy archived at **PRC-comp, 2023**).

To complement GNM, we used a hybrid molecular simulations approach that integrates global elastic modes with atomic-scale information derived from all-atom simulations. Several such approaches have emerged in recent years (e.g. collective MD [CoMD], Perturb-Scan-Pull, eBDIMS, H-REMD) (*Krieger et al., 2020*; *Gur et al., 2013*; *Jalalypour et al., 2020*; *Kandzia et al., 2019*) with the aim of finding a 'sweet spot' between the precision of whole atom simulations and the computational efficacy of coarse-grained elastic networks. The novelty of our approach, ANM-LD, is that it integrates a low-resolution α-carbon-based ANM with stochastic all-atom implicit solvent LD simulations (*Brünger et al., 1984*). ANM-LD reflects only internal dynamic modes, free of any external bias. More importantly, ANM-LD provides experimentally testable predictions, and has been successfully used to compute conformational trajectories of proteins (*Acar et al., 2020*; *Yang et al., 2018*). The code for ANM-LD is freely available on GitHub, (copy archived at *PRC-comp, 2023*).

By combining ANM-LD and GNM-TE, we present here the allosteric network of CFTR. The results align very well with experimental and clinical data, disclose the allosteric mechanism of CFTR, and identify a novel allosteric hotspot that may be used for therapeutic intervention.

## Results
### TE calculations identify allosteric hotspots in CFTR
First, we performed GNM-TE calculations on non-phosphorylated ATP-free human CFTR (PDB ID 5UAK, *Liu et al., 2017b*) using the 10 slowest GNM 'modes'. As explained in the Materials and methods section and elsewhere (*Acar et al., 2020*; *Haliloglu et al., 1997*), GNM decomposes a

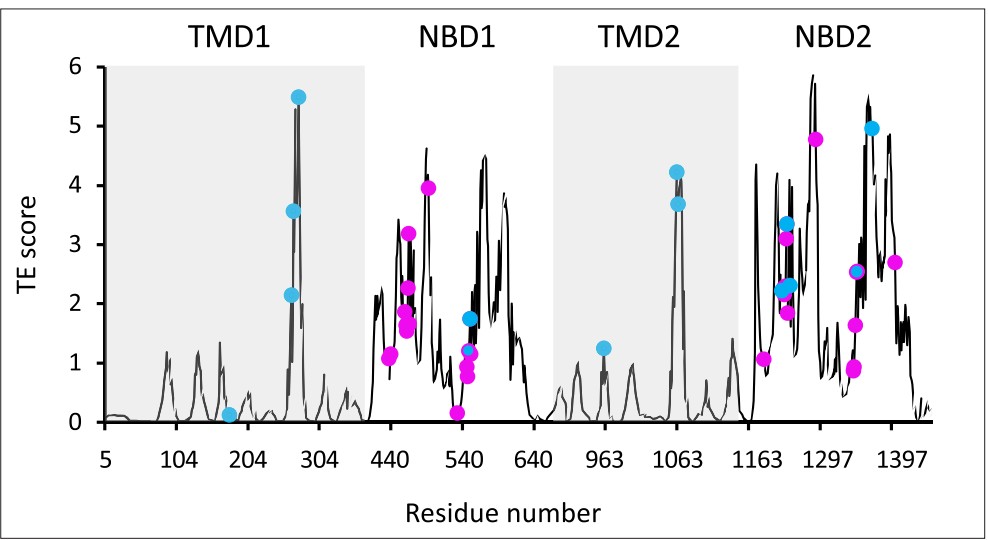

**Figure 1.** Functional sites of cystic fibrosis transmembrane conductance regulator (CFTR) segregate to allosteric hotspots. Shown is the amount of information (transfer entropy [TE] score) transmitted by each residue of dephosphorylated ATP-free human CFTR (PDB ID 5UAK) calculated using the 10 most collective Gaussian network model (GNM) modes (solid black trace). The positions of the 14 functionally important residues (see text for details) and of 30 ATP-binding residues are shown as cyan and magenta spheres, respectively.

The online version of this article includes the following figure supplement(s) for figure 1:

**Figure supplement 1.** Eigenvalue decays as a function of mode number, shown for ATP-free (orange) and ATP-bound (blue) human cystic fibrosis transmembrane conductance regulator (CFTR) (PDB ID 5UAK and 6MSM, respectively).

**Figure supplement 2.** Sensitivity of the results to $R_{cut}$.

**Figure supplement 3.** Co-localization of the Gaussian network model-transfer entropy (GNM-TE) peaks with the hinges identified by GNM.

**Figure supplement 4.** Correlation between allosteric peaks and positions of the functionally essential residues.

**Figure supplement 5.** Correlation between allosteric peaks and ATP-binding residues.

**Figure supplement 6.** Correlation between allosteric peaks and positions of disease-causing mutations.

complex motion (such as a conformational change, or a person walking) to discrete dynamic building blocks, historically termed GNM 'modes', the ensemble of which comprise the complex motion. These include modes of motion that involve large parts of the protein and lead to major conformational changes (termed 'slow modes'). The 'slow modes' often display the greatest degree of collective movements. Modes that involve local conformational adjustments and a smaller number of residues and display fewer collective movements are considered 'fast modes'. We expected that the 10 slowest modes, those comprising the largest of motions, will be the main allosteric effectors, and their ensemble will suffice to capture the overall dynamics of CFTR. Indeed, Eigenvalue analysis demonstrated that the 10 slowest modes of CFTR encompass its complete dynamic spectrum (*Figure 1—figure supplement 1*).

*Figure 1* shows the residues identified as the main allosteric sources in ATP-free human CFTR, obtained by using an interaction cutoff distance ($R_{cut}$) of 10 Å. Very similar results were obtained when $R_{cut}$ of 7 Å was used instead (*Figure 1—figure supplement 2*), suggesting that the results are not very sensitive to this parameter. The residues that GNM-TE identified as main allosteric determinants showed partial overlap with the hinge residues identified by GNM: Of the 60 residues that serve as main allosteric sources, 22 were located within 4 Å of a hinge residue, and this co-localization was highly statistically significant (*Figure 1—figure supplement 3*). This is perhaps not unexpected, as hinge residues likely play an important role in transmitting the strain energy induced by binding of allosteric ligands.

As GNM-based TE has not been extensively used to study membrane proteins as large and complex as CFTR, we began by evaluating its utility in extracting meaningful functional information. Allosteric residues often coincide with functional epitopes, i.e., residues that are involved in ligand binding, catalysis, or serve as conformational hinges (*Liu and Nussinov, 2016*; *Wodak et al., 2019*). Therefore, we first evaluated the performance of GNM-based TE by examining its ability to identify such sites of functional importance. For this, we first compared the locations of the GNM-TE identified allosteric sources to the positions of 14 residues that have been experimentally demonstrated to be directly responsible for the function of CFTR. This subset of 14 residues includes the most common CFTR gating mutations (G178, S549, G551, G970, G1244, S1251, S1255, G1349), residues of the intracellular loops that are essential for conformational changes (M265, N268, Y275, W1063, L1065), and the NBD2 Walker B aspartate D1370 which is essential for ATP hydrolysis (*Liu et al., 2017b*; *Csanády et al., 2019*; *Sorum et al., 2015*; *Sorum et al., 2017*). As shown in *Figure 1* we observed that these 14 residues tightly co-localize with the main allosteric peaks. To assess the statistical significance of this co-segregation, we compared it to a random identification of residues of functional importance, as follows: GNM-TE identified 60 residues (out of 1480) that are the main allosteric sources (i.e. allosteric peaks) in dephosphorylated ATP-free CFTR (*Figure 1*). Accordingly, we generated 100,000 sets of 60 randomized positions and counted how many times the positions of the functionally essential residues were correctly predicted using this random allocation. As shown in *Figure 1—figure supplement 4A*, the random allocation has a mean value of 0.7 correct guesses while the GNM-TE-based identification has a mean value of 8 ($p=9 \times e^{-13}$). Next, we expanded this analysis to include not only exact matches (between the random or GNM-TE assignment of functional sites and their real location) but also first- or second-coordination sphere interactions (i.e. peaks that are located <4 Å or <7 Å from the functionally essential residues). Using these cutoffs, we also observed a highly significant correlation between the location of the GNM-TE peaks and that of the functionally essential residues (*Figure 1—figure supplement 4B–C*). We then compiled a second set of reference residues, one which includes all the residues that provide the first ATP coordination sphere in ATP-bound CFTR (*Zhang et al., 2018*). We reasoned that since CFTR gating is controlled by ATP binding and hydrolysis, these residues (that are allosteric by definition) should be readily identified by the GNM-TE analysis. Indeed, as shown in *Figure 1* the ATP residues also closely segregated to the allosteric peaks identified by GNM-TE, and this co-segregation outperformed the one obtained via a random allocation of allosteric peaks (*Figure 1—figure supplement 5*). This co-segregation was observed whether we considered only exact matches (*Figure 1—figure supplement 5A*), or whether we included also first- or second-coordination sphere interactions (*Figure 1—figure supplement 5B and C*, respectively). Next, we considered a more challenging reference set of residues, one which includes the 93 CFTR missense point mutations that lead to CF (*Cftr2.org, 2011*). This reference set of residues holds some inherent limitations: many of the mutations that lead to CF have no direct functional role per se, rather,

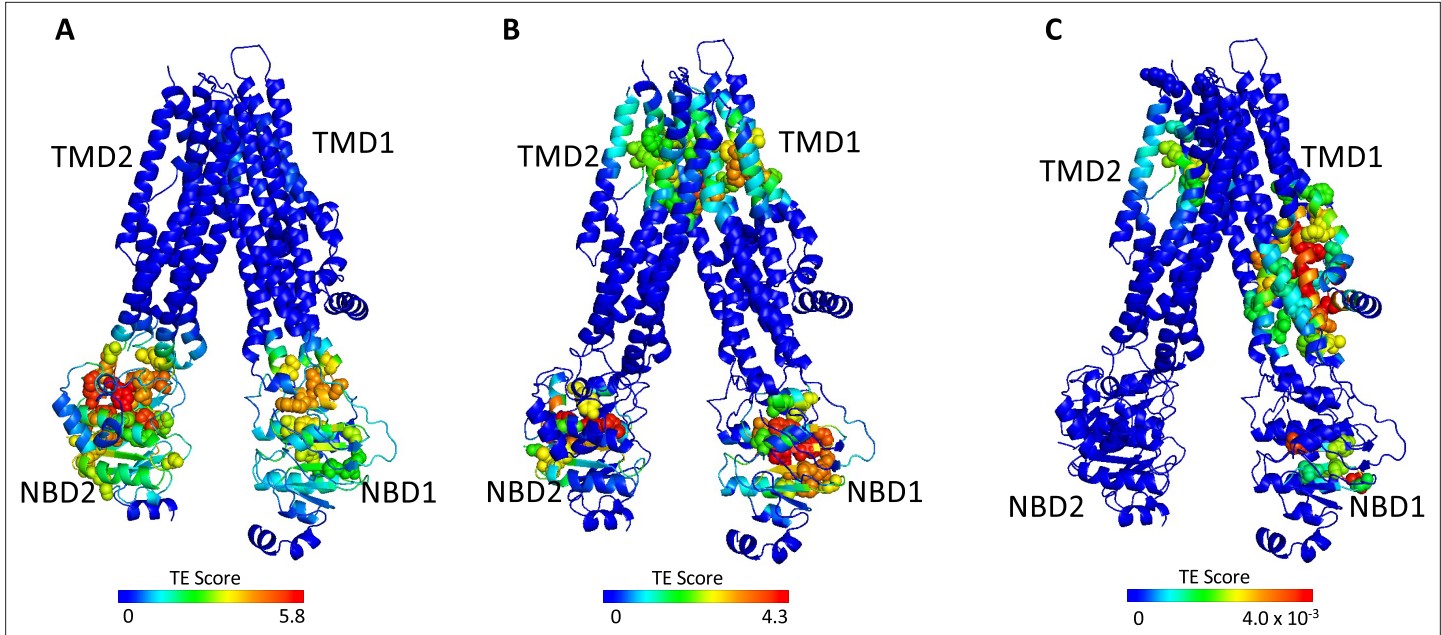

**Figure 2.** Identification of allosteric hotspots in dephosphorylated ATP-free cystic fibrosis transmembrane conductance regulator (CFTR). A cartoon representation of dephosphorylated ATP-free human CFTR (PDB ID 5UAK), where each residue is colored according to the amount of information it transmits (transfer entropy [TE] score), expressed as transfer entropy × collectivity (TECol) calculated using the 10 most collective Gaussian network model (GNM) modes (**A**), or upon removal of the first (**B**) or first and second (**C**) most collective GNM modes. Red and blue colors represent high and low levels of information output, respectively.

they cause CF due to compromised mRNA synthesis/stability, protein processing, or protein targeting (**Chen et al., 2021**). The inclusion of such residues in the benchmark dataset will lead to apparent false-negative identifications of the GNM-TE analysis. In addition, it is likely that not all the allosteric residues have been naturally mutated, leading to false-positive identifications. Nevertheless, despite these limitations, we expected to see some degree of co-localization between the identified allosteric hotspots and the disease-causing mutations. From the 93 mutations listed in the cfrt2 database (**Cftr2.org, 2011**), we removed the cluster of seven N-terminal residues (S13, L15, W60, E56, E60, P67, G85) that are known to be involved in interactions with cellular filamins or ER retention (**Thelin et al., 2007**). As shown in **Figure 1—figure supplement 6A** we observed considerable association between the positions of 86 disease-causing mutations and the allosteric peaks identified by GNM-TE. To test the statistical significance of this association, we generated 100,000 sets of randomly positioned 'allosteric peaks'. As shown in **Figure 1—figure supplement 6B**, the random distribution is roughly normal with a mean of 4.5 incidents of exact matches. As shown in **Figure 1—figure supplement 6B**, the GNM-TE analysis is significantly different from the random distribution, with a mean of 9 exact matches. As observed with the previous two reference set of residues (**Figure 1—figure supplements 4 and 5**), the co-segregation between the allosteric peaks identified by GNM-TE and the positions of the disease-causing mutations was also observed when we included also first- and second-coordination sphere interactions (**Figure 1—figure supplement 6C and D**, respectively). In further support of the biological relevance of the GNM-TE calculations, and despite the structural similarity between the two NBDs, a clear asymmetry can be observed between them, with NBD2 which harbors the catalytic Walker B glutamate providing the higher allosteric signaling (**Figures 1 and 2A**). This observed asymmetry is in line with a recent single molecule conformational study where the degenerate and catalytic ATP sites were shown to asymmetrically affect NBD dimerization and gating (**Levring et al., 2023**). That GNM-TE identifies this asymmetry is quite remarkable, as it is based solely on the positions of the α-carbons and does not consider the side chains. Taken together, these results (**Figures 1 and 2A**, **Figure 1—figure supplements 2 and 4**, **Figure 1—figure supplements 5 and 6**) suggest that GNM-based TE can provide meaningful functional information on CFTR.

As shown in **Figures 1 and 2A**, the entropy sources residues (i.e. residues that drive allostery) are found predominantly on the NBDs and are largely absent from the TMDs. The only TMD residues that

serve as entropic sources are those found at or near intracellular loops 2 and 3 (*Figure 2A*). Notably, these observations were made for de-phosphorylated ATP-free CFTR. This demonstrates that even when the NBDs are completely separated and the composite ATP-binding sites are yet to form, the dynamic infrastructure for nucleotide binding is already present, poised for binding of the ligand. This pre-existing dynamic infrastructure likely facilitates ATP binding and will come into full play at a later stage of the conformational change, following phosphorylation and binding of ATP. Interestingly, some of the main entropy peaks (i.e. entropy sources) located in the NBDs are slightly shifted from the ATP-binding residues (*Figure 1*) and correspond to adjacent residues. This shift disappears once ATP is bound, and the entropy source role is transferred from the neighboring residues to the ATP-binding residues themselves (see next results section dealing with the effects of ATP binding).

The results shown in *Figure 2A* were obtained using the 10 slowest GNM modes. However, during the calculations the entropic contribution of the slowest modes may overshadow that of the faster (less collective) ones, which may nevertheless have important functional roles (for more details, see *Altintel et al., 2022*). To circumvent this problem, one may repeat the calculations while omitting the slowest GNM modes, enabling the detection of latent allosteric interactions that may otherwise be hidden in the global fluctuations. For ATP-free human CFTR, the greatest effect was observed upon removal of the first and second slowest modes: Upon removal of the slowest GNM mode, a new cluster of residues that serve as allosteric sources was revealed (*Figure 2B*). These residues, which are distributed roughly symmetrically in the TMDs, include the putative gating residues (e.g. P99, T338, S341, *Liu et al., 2017b*) and additional residues that line the ion permeation pathway. The residues that form the ivacaftor binding site (*Liu et al., 2019*) were also revealed as entropy sources by removal of the slowest GNM mode (*Figure 2B*), demonstrating the utility of this approach. Removal of the next slowest GNM mode reveals another cluster of TMD residues that serve as entropy/allostery sources. Unlike the above-described symmetric cluster, this cluster is highly asymmetric, and is comprised solely of residues in TMD1 (*Figure 2C*). Interestingly, this cluster of residues is located at and just above the future site for docking of the regulatory domain (*Bozoky et al., 2013*; *Chappe et al., 2005* and see also later in *Figure 3C–D*).

The above analysis revealed two latent entropy hotspots: one surrounding the ion permeation pathway (*Figure 2B*) and the other adjacent to the docking site of the regulatory domain (*Figure 2C*). Both sites were hidden by the more dominant global fluctuations of the protein (*Figure 2A*). Their identification in this conformation, which is non-conducting and where the R domain is still wedged between the NBDs (*Liu et al., 2017b*), highlights once again the pre-existence of a dynamic infrastructure which will assume a central role following phosphorylation and repositioning of the R domain.

## ATP binding and phosphorylation rewires and focuses the allosteric connectivity of CFTR

To study the effects of ATP binding and phosphorylation, we conducted TE analysis of phosphorylated ATP-bound CFTR (PDB ID 6MSM, *Zhang et al., 2018*). We observed that phosphorylation and binding of ATP greatly rewires the overall allosteric connectivity in CFTR (*Figure 3—figure supplement 1*, and compare *Figures 1 and 3A*): residues that serve as entropy sources are now found in all the domains (TMD1, NBD1, TMD2, NBD2) and the entropy peaks are sharp, sharper than those observed for the unphosphorylated ATP-free form (compare *Figures 1 and 3A*), indicating that information is originating from distinct residues/cluster of residues. This suggests that phosphorylation and binding of ATP focuses the dynamics of CFTR, which pre-existed in a more diffuse manner in the unphosphorylated ATP-free state. This observation suggests that this conformation is the preferred template/target for structure-guided development of allosteric modulators.

As discussed above and shown in *Figure 1*, in the ATP-free state the entropy peaks at the NBDs are slightly shifted from the ATP-binding residues. This offset largely disappears upon binding of ATP, and the entropy peaks in the NBDs now closely match the location of the ATP-binding residues (compare *Figures 1 and 3A*). In the TMDs, the residues that serve as main entropy sources are those that surround the ion permeation channel (*Figure 3B*). This means that upon phosphorylation and binding of ATP the gating and pore lining residues assume a more dominant role in allosteric signaling. Notably, these transmitting TMD residues were identified as latent entropy sources in the unphosphorylated ATP-free state (*Figure 2B*). This result again highlights that the underlying dynamics in CFTR pre-exist, prior to phosphorylation and ATP binding. To test if the effect of ATP binging is specific to human CFTR and/or

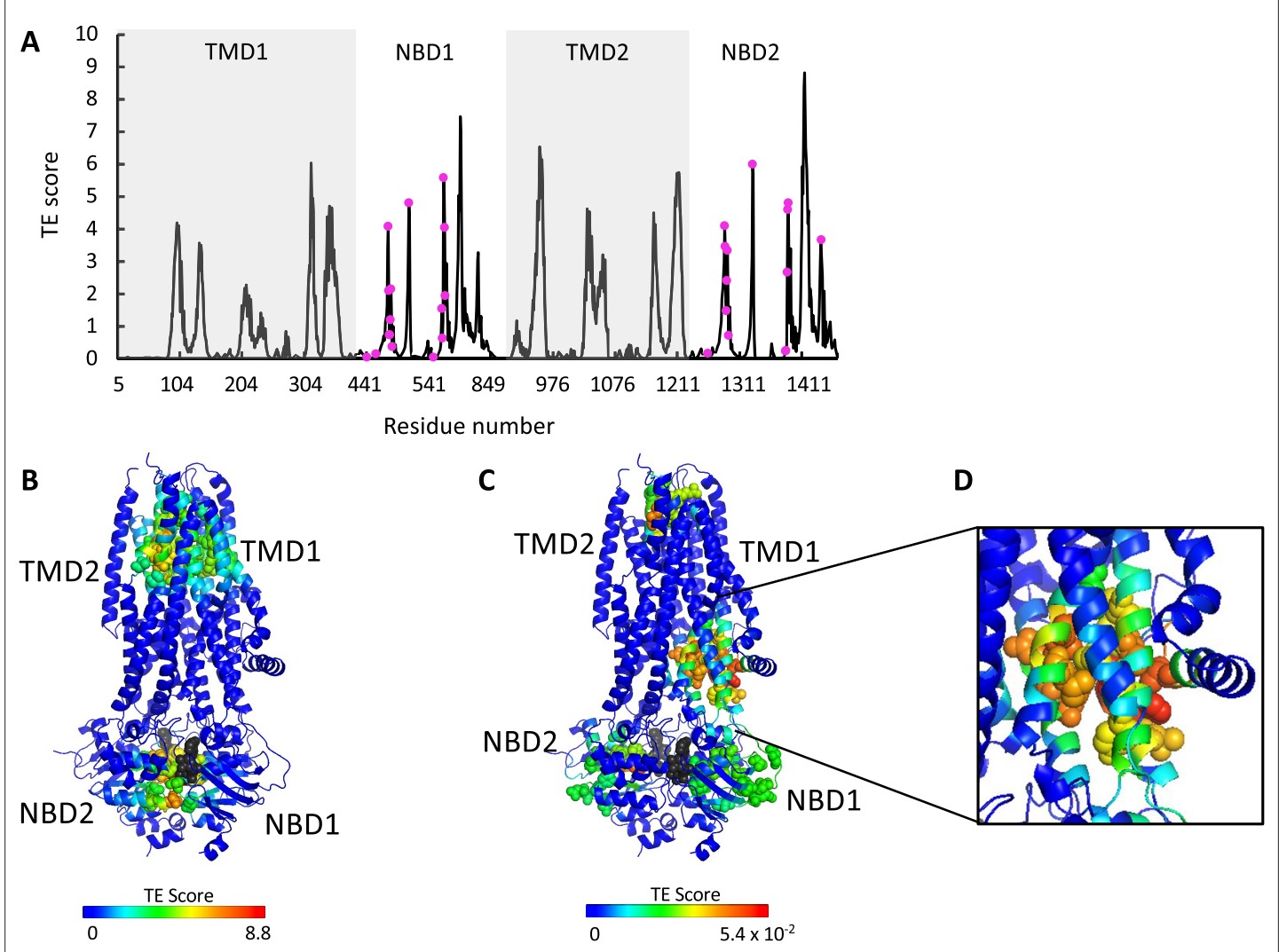

**Figure 3.** ATP binding and phosphorylation rewires and focuses the allosteric connectivity of cystic fibrosis transmembrane conductance regulator (CFTR). (**A**) Shown is the amount of information transmitted (transfer entropy [TE] score) by each residue of phosphorylated ATP-bound human CFTR (PDB ID 6MSM) calculated using the 10 most collective Gaussian network model (GNM) modes (solid black trace). The positions of the ATP-binding residues are shown as magenta spheres. (**B, C**) A cartoon representation of phosphorylated ATP-bound human CFTR (PDB ID 6MSM), where each residue is colored according to the amount of information it transmits (TE score), calculated using the 10 most collective GNM modes (**B**), or upon removal of the first most collective GNM mode (**C**). Red and blue colors represent high and low levels of information output, respectively, and the ATP molecules are shown as black/gray spheres. (**D**) A magnified view of the allosteric cluster that is adjacent to the docking site of the R domain.

The online version of this article includes the following figure supplement(s) for figure 3:

**Figure supplement 1.** Phosphorylation and ATP binding rewires the allosteric connectivity in cystic fibrosis transmembrane conductance regulator (CFTR).

**Figure supplement 2.** Conservation of the effect of ATP.

is a result of the specific experimental conditions in which its ATP-free and ATP-bound structures were determined (***Liu et al., 2017b***; ***Zhang et al., 2018***), we conducted similar calculations with the corresponding conformations of zebrafish CFTR (***Zhang and Chen, 2016***; ***Zhang et al., 2017***). As shown in ***Figure 3—figure supplement 2***, binding of ATP had a very similar effect in both CFTR homologues, suggesting that the allosteric effects of ATP binding are robust and conserved.

As explained above (see analysis of the ATP-free conformation), to identify latent entropy sources we repeated the TE calculations for the phosphorylated ATP-bound state after removing the two slowest modes. Several striking features were revealed by these calculations (***Figure 3C***). First, the entropy sources in the NBDs have now migrated from the ATP-binding sites at the dimer interface

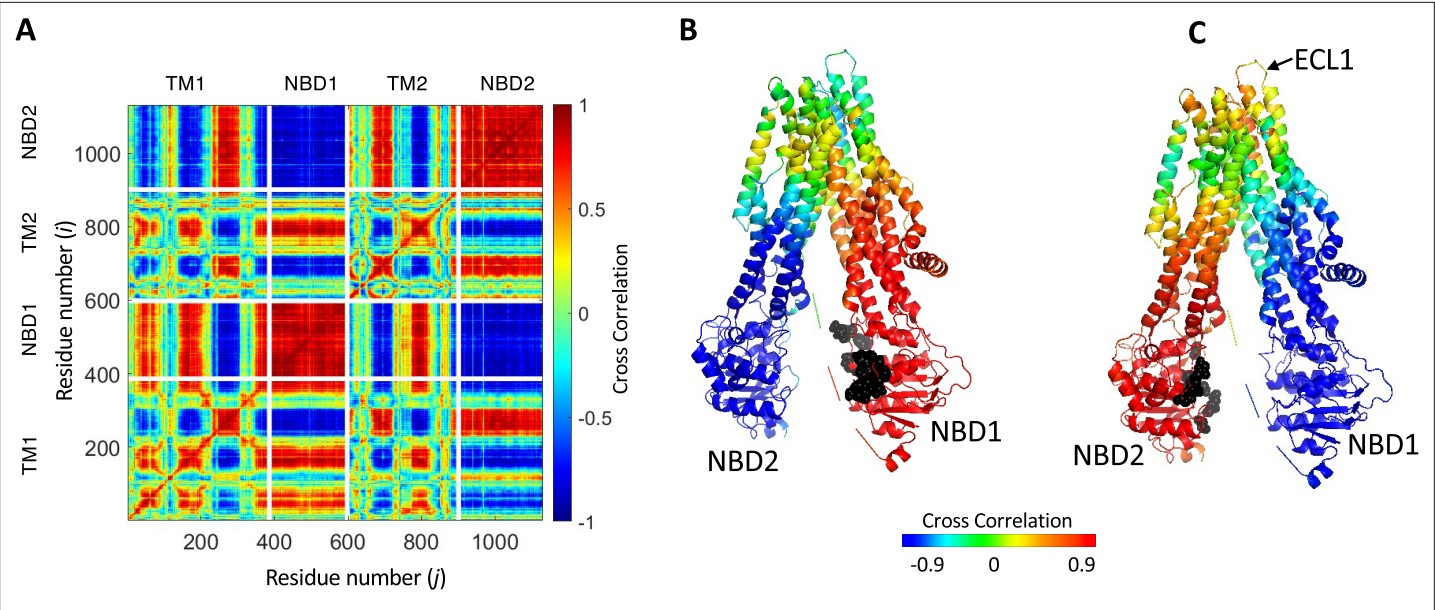

**Figure 4.** Synchronized movements of residues and domains of cystic fibrosis transmembrane conductance regulator (CFTR). (**A**) Shown is a 2D map of the dynamic cross-correlations between all residues of CFTR during the conformational transiting from the dephosphorylated ATP-free state to the phosphorylated ATP-bound state (PDB ID 5UAK and 6MSM, respectively). Red colors indicate strong positive correlation, meaning that the residues move in parallel vectors in space and time, blue colors indicate negative correlations. Domain boundaries are indicated by white lines. (**B, C**) Cartoon representation of CFTR where each residue is colored according to its dynamic cross-correlations with the ATP-binding residues (black spheres) of nucleotide binding domain (NBD)1 (**B**) or NBD2 (**C**).

toward the intracellular loops, perfectly positioned to transmit the signal to the TMDs. Second, the TMD allosteric peaks surrounding the gating residues and the ion permeation pathway have migrated toward the extracellular exit, as if to prepare for ion exit. Lastly, a cluster of residues that serve as strong entropy sources appeared in the TMDs in an asymmetric manner (*Figure 3C–D*). These residues are found exclusively in TMD1 and comprise or are adjacent to the site of interaction of the regulatory (R) domain (*Bozoky et al., 2013*; *Chappe et al., 2005*). This observation suggests that the role of the R domain may not be limited to steric obstruction of NBD dimer formation in the unphosphorylated ATP-free state. The presence of this cluster of information-transmitting residues in the vicinity of the docking site of the R domain may explain previous observations that deletion of the R domain leads to fewer channel opening events, and that exogenous addition of the phosphorylated R domain peptide increases channel open probability of CFTR construct deleted of the R domain (*Winter and Welsh, 1997*; *Ma et al., 1997*).

## Concerted movements in CFTR

The TE analysis presented above provides a measure of information transfer between residues and domains. However, it gives no information of the physical nature of the conformational changes that are involved. To complement the GNM-TE calculations, we used ANM-LD, a molecular simulations approach which computes the trajectory of a conformational change between two known states of a protein (*Atilgan et al., 2001*; *Brünger et al., 1984*). We simulated the transition between non-phosphorylated ATP-free and phosphorylated ATP-bound human CFTR (*Liu et al., 2017b*; *Zhang et al., 2018*). To understand the allosteric connectivity that underlies this conformational change, we analyzed the degree of correlated movements between all CFTR residues. At the basis of this analysis is the assumption that residues that are allosterically connected will move in synchrony in both space and time. An extreme example of such synchronization is the perfect correlation of each residue with itself (diagonal in *Figure 4A*). Similarly, two residues that are sequential in the amino acid sequence will also display high correlation. However, when highly synchronized movements are observed for residues or domains that are distant in sequence and in 3D space, this likely represents allosteric connectivity.

The 2D cross-correlation map obtained for the transition between non-phosphorylated ATP-free and phosphorylated ATP-bound human CFTR is shown in *Figure 4A*. In this map red colors indicate high positive correlations and blue colors high negative correlations. Red colors thus indicate residues that move in parallel vectors in space and time, and blue colors indicate residues that move in anti-parallel vectors in space and time. The map is dominated by large and continuous patches of red (positive correlation) and blue (negative correlation) areas and is very similar to the 2D cross-correlation map calculated for PglK (*Acar et al., 2020*), a lipid-linked oligosaccharide flippase that adopts the canonical ABC exporter 3D fold (*Perez et al., 2015*). These observations suggest that, as inferred from the cryo-EM structures of CFTR (*Liu et al., 2017b*; *Zhang et al., 2018*), global rigid body motions with a relatively small number of hinges underlie this conformational change, and that the conformational changes of CFTR are similar to those of canonical ABC exporters. The two NBDs display asymmetric allosteric connectivity: NBD1 shows a tighter correlation with the residues that surround the permeation pathway (*Figure 4B*), while NBD2 (harboring the catalytic Walker B glutamate) is more strongly correlated with ECL1 (extracellular loop 1) (*Figure 4C*) which is known to stabilize the conducting state of the channel (*Infield et al., 2016*).

## Sequence of allosteric transduction in CFTR

The ANM-LD simulations offer an opportunity to examine mechanistic features that are difficult to determine experimentally. One such feature is the pre-equilibrium kinetics of allosteric interactions: if residues A and B are allosterically connected, does the conformational change of A follow that of B, or vice versa? To investigate the sequence of allosteric transduction in CFTR we re-analyzed the simulations while imposing a time delay between the conformational changes of allosterically coupled residues (or domains) of interest. For any two allosterically coupled residues, if we define that the conformational change in A occurs before that of B, and still observe a positive correlation in the 2D correlations map, this means that the conformational change of A precedes that of B. If the correlation is lost, this means that B likely precedes A, which can be confirmed by imposing the opposite time delay (i.e. imposing that the change in B precedes) and verifying that the positive correlation is again observed (see Materials and methods for a mathematical expression of these relations). This approach can be used to track the entire route of allosteric signal transduction as detailed below.

Rate-equilibrium free-energy relationship analysis and single molecule FRET studies both suggest that the allosteric signal originates at the NBDs and propagates to the TMDs (*Sorum et al., 2015*; *Levring et al., 2023*). Our results agree with these observations: in parallel simulations we repeatedly observed that the conformational change from the dephosphorylated ATP-free state to the phosphorylated ATP-bound one begins with the movement of the NBDs. In addition, the TE analysis of the inward-facing ATP-free state demonstrated that in this conformation the allosteric peaks are located almost exclusively at the NBDs (*Figure 1*). Of the two NBDs, the catalytic NBD2 seems to provide the greater allosteric input (*Figure 1* and *Figure 2A* ) and therefore as detailed below we began tracing the allosteric signal at NBD2. As shown above (*Figure 4C*), in the absence of a time delay the movements of NBD2, ICL2/3, TMD2 (including TM8 up to its hinge), and ECL1 are highly synchronized. However, this correlation persists only if NBD2 leads this motion and ICL2/3, TMD2/TM8, and ECL1 follow (*Figure 5—figure supplement 1* green rectangles, *Figure 5* step I). This observation strongly suggests that the movement of NBD2 is the leading event and those of ICL2/3, TMD2/TM8, and ECL1 follow, and that the allosteric signal propagates from NBD2 to ICL2/3, TMD2/TM8, and ECL1 (*Figure 5* center, green arrows). Notably, based on their recent single molecule studies, Levring et al. proposed similar cause-and-effect relations between the catalytic ATP site and TM8 (*Levring et al., 2023*).

From ECL1 the allosteric signal propagates to NBD1: The movements of these two domains are highly correlated, but only if ECL1 leads and NBD1 follows (*Figure 5—figure supplement 1* magenta rectangles, *Figure 5* step II, and *Figure 5* center, magenta arrow). Similar considerations suggest that from NBD1 the allosteric signal propagates to the gating residues and to the TM helices that surround the permeation pathway: The movements of these domains are only synchronized if the former leads and the latter follows (*Figure 5—figure supplement 1* blue rectangles, *Figure 5* step III, and *Figure 5* center, blue arrow). In turn, the movements of the gating residues and the permeation pathway TM helices are correlated with those of NBD2, but only if the TM helices lead and NBD2 follows (*Figure 5—figure supplement 1* orange rectangles, *Figure 5* Step IV). We therefore conclude

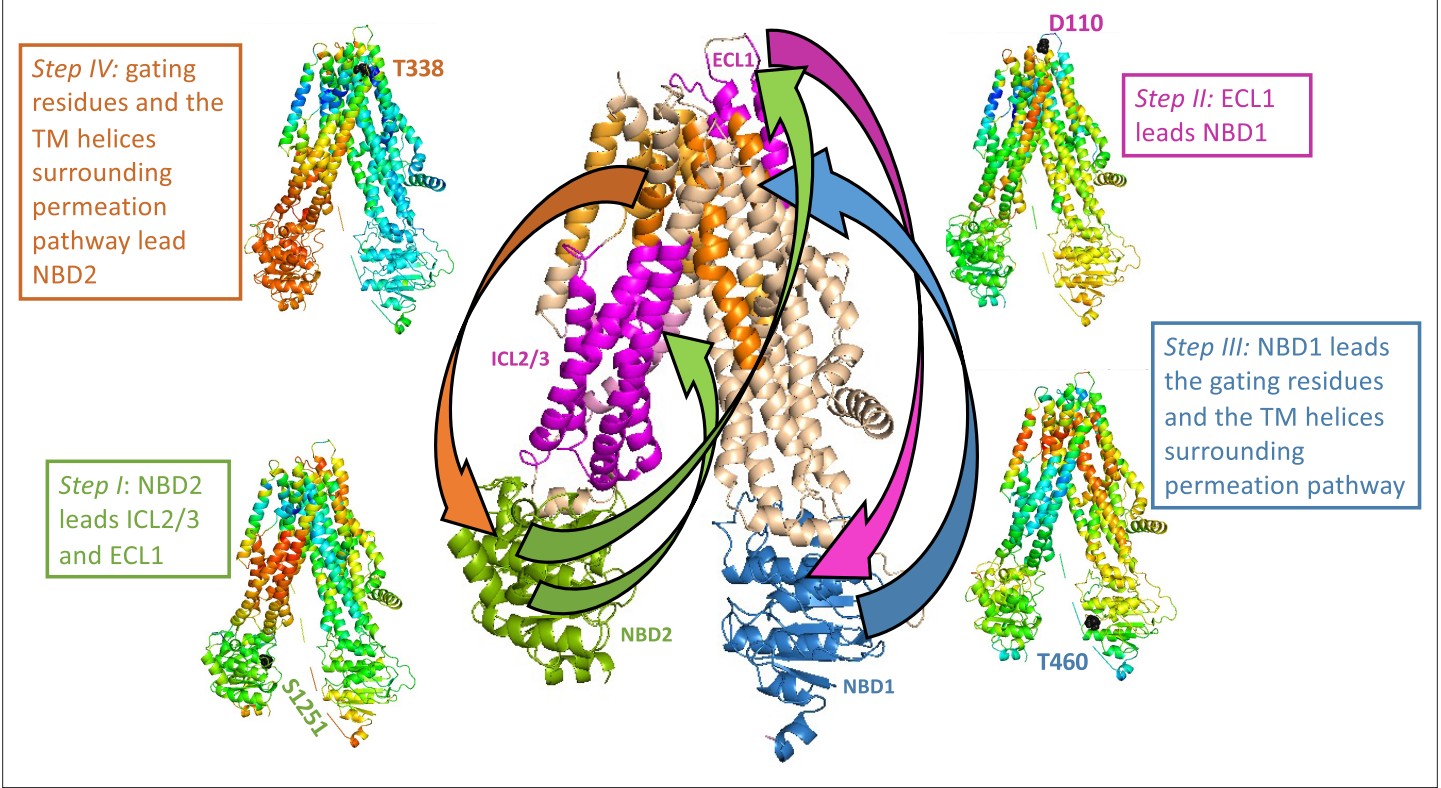

**Figure 5.** Trajectory and sequence of allosteric transduction in cystic fibrosis transmembrane conductance regulator (CFTR). Dynamic cross-correlations were calculated using a time delay ($\tau$) of 16 cycles out of 50 required to complete the conformational transition between the dephosphorylated ATP-free and phosphorylated ATP-bound states. Red and blue colors indicate high and low correlations, respectively. *Steps I–IV:* Each residue is colored according to the degree of the correlation of its movement at time ($t + \tau$) relative to the movement at time ($t$) of S1251 of nucleotide binding domain (NBD)2 (step I), D110 of ECL1 (step II), T460 of NBD1 (step III), and the gating residue T338 (step IV). *Center (large) panel:* Shown is a summary of the allosteric trajectory, originating from NBD2 (green) to ICL2/3 and ECL1 (magenta), from ECL1 to NBD1 (blue), from NBD1 to the permeation pathway helices (orange), and finally back to NBD2.

The online version of this article includes the following figure supplement(s) for figure 5:

**Figure supplement 1.** Directionality of allosteric transduction in cystic fibrosis transmembrane conductance regulator (CFTR).

that the allosteric signal propagates from the permeation pathway to NBD2, thus completing the cycle of allosteric transduction (*Figure 5* center, orange arrow).

## Modulation of CFTR by drugs

In the past decade, the development of small-molecule CFTR modulators revolutionized CF treatment and led to a dramatic reduction in patient morbidity and increased life expectancy (*Clancy, 2018*). There are currently four FDA-approved drugs that are used to treat CF: The type I correctors lumacaftor (VX-809) and tezacaftor (VX-661) stabilize mutant CFTR variants (predominantly the most common ΔF508 mutation) reducing premature protein degradation and improving surface presentation of the mature channel (*Van Goor et al., 2011*; *Fiedorczuk and Chen, 2022b*). The potentiator ivacaftor (VX-770) enhances the activity of conductance-defective mutants by increasing the open probability (Po) of the channel (*Levring et al., 2023*; *Liu et al., 2019*; *Eckford et al., 2012*), and the dual-function type III corrector elexacaftor (VX-445) stabilizes unstable variants and increases channel open probability (*Shaughnessy et al., 2021*; *Veit et al., 2021*). The recently published structures of WT and ΔF508CFTR bound to various drugs (*Fiedorczuk and Chen, 2022a*) enabled us to investigate the allosteric effects of the drugs. First, we conducted GNM-TE analysis on ΔF508CFTR bound to Trikafta (PDB ID 8EIQ, *Fiedorczuk and Chen, 2022a*), a trivalent drug which combines ivacaftor, tezacaftor, and elexacaftor and is currently the most advanced CF treatment. As shown in *Figure 6A*, the type I correctors lumacaftor and tezacaftor bind at 'valleys', i.e., to residues that are not allosteric sources. These results are consistent with the mechanism of these drugs which improve the function

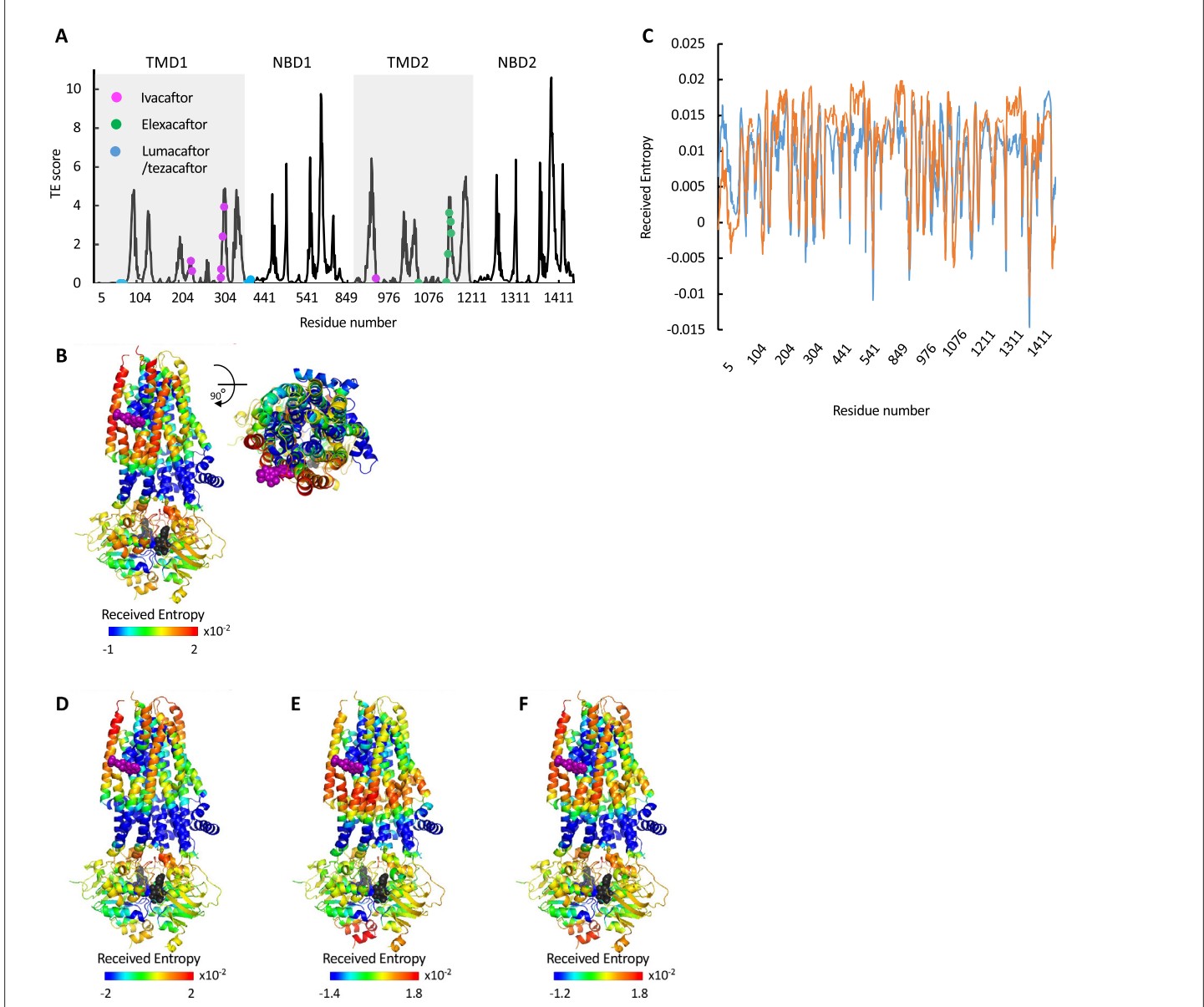

**Figure 6.** Allosteric modulation of cystic fibrosis transmembrane conductance regulator (CFTR) by drugs. (**A**) Shown is the amount of information TRANSMITTED (transfer entropy [TE] score) by each residue of CFTR ΔF508 bound to Trikafta (PDB ID 8EIQ) calculated using the 10 most collective Gaussian network model (GNM) modes (solid black trace). The binding sites of ivacaftor, elexacaftor, and lumacaftor/tezacaftor are shown as magenta, green, or cyan spheres, respectively. (**B**) Cartoon representations of ivacaftor bound CFTR (PDB ID 6O2P), with ivacaftor and ATP shown as magenta and black/gray spheres, respectively. Residues are colored according to the amount of entropy RECEIVED from the ivacaftor binding residue F312 (**C**) Shown is the amount of entropy RECEIVED by each residue of CFTR from the ATP-binding residue G551 (blue trace) or the ivacaftor binding residue F312 (orange trace). (**D–F**) Same as B, only the residues are colored according to the amount of entropy RECEIVED from the gating residue F337 (**D**), the ATP-binding residue G551 (**E**), and the sum of the entropy received from both G551 and F337 (**F**).

of CFTR by lowering its ΔG$_{(folding)}$ rather than by modulating its activity (*Fiedorczuk and Chen, 2022b*). In contrast, the residues that comprise the binding sites for ivacaftor and elexacaftor also include ones that serve as main allosteric sources (*Figure 6A*), alluding to their role as allosteric modulators.

In CFTR, channel opening depends on phosphorylation of the regulatory domain and subsequent binding of ATP by the NBDs. Accordingly, mutations that interfere with binding of ATP reduce channel opening probability and result in CF. The potentiator drug ivacaftor is used to treat CF patients who carry mutations in the NBDs (most commonly the G551D mutation at the highly conserved LSGGQ ATP binding/signature motif) (*Liu et al., 2019*; *Eckford et al., 2012*; *Hadida et al., 2014*; *Yu et al.,*

*2012*). Ivacaftor increases channel opening probability despite impaired ATP binding and hydrolysis. The molecular basis for the beneficial effects of ivacaftor is not fully understood, especially since it does not bind in the vicinity of either the ATP- or ion-coordinating residues but rather peripherally at the protein-lipid interface (*Liu et al., 2019*). In addition, unlike the folding correcting drugs, ivacaftor does not induce a significant conformational change in ΔF508 CFTR (*Fiedorczuk and Chen, 2022a*). To better understand the mechanism underlying the CFTR's modulation by ivacaftor, we mapped the residues that receive information from the ivacaftor binding site. Surprisingly, the gating residues and those that line the permeation pathway were not among the residues that receive information from the drug binding site (*Figure 6B*), leaving the molecular basis of the drug's action unclear. Thus, in attempt to resolve this issue, we compared the profile of the residues that receive information from the ivacaftor binding site to those that receive information from the ATP-binding sites. As shown in *Figure 6C*, the two profiles showed remarkable overlap: The ivacaftor docking site and the ATP-binding site send information to a very similar set of residues (Pearson's correlation coefficient of 0.83). These observations suggest that ivacaftor increases the opening probability not by directly affecting the gating residues but rather indirectly by mimicking the allosteric signaling evoked by ATP binding. Further insight for the mechanism of action of ivacaftor can be obtained from the structural representation of the allosteric signaling shown in *Figure 6D–F*. The gating region (e.g. gating residue F337) (*Zhang et al., 2018*) sends information mostly to the extracellular side of TM helices 2 and 11 and to the cytoplasmic side of helices 3, 4, 7 including intracellular loops 2 and 3 (*Figure 6D*). In comparison, G551 of the ATP-binding signature motif (mutations in which are treated with ivacaftor) sends information predominantly to the intracellular side of TM helices 3, 4, 7, 8 and to the C-terminal intracellular helix of NBD2 (*Figure 6E*). Superimposition of these effects, which represents the combined allosteric signaling of the ion-coordinating and ATP-binding residues (shown in *Figure 6F*), is very similar to the allosteric signaling sent by the ivacaftor binding residue F312 (*Figure 6B*). This result suggests that ivacaftor exerts its potentiating effect by mimicking the combined allosteric output of the ATP-binding and gating residues, compensating for any potential allosteric transduction short circuit caused by mutations.

## Discussion – the allosteric mechanism of CFTR

Allostery in proteins is most often studied through the lenses of the classical KNF or MWC models, and more recently the dynamic mechanism (see Introduction and *Monod et al., 1965*; *Motlagh et al., 2014*; *Nussinov and Tsai, 2015*; *Ribeiro and Ortiz, 2016*). The results presented herein suggest that none of these models can singly explain the allosteric mechanism of CFTR.

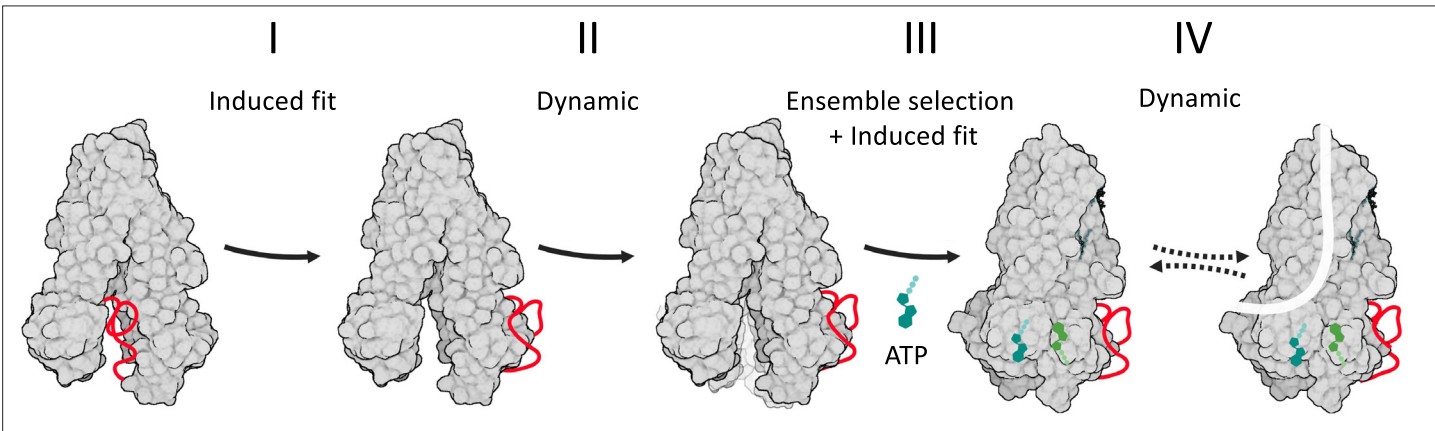

**Figure 7.** Model of cystic fibrosis transmembrane conductance regulator (CFTR's) conformational changes and their allosteric control. CFTR is shown in a gray space-filling model, the regulatory (R) domain as a red ribbon, and ATP in shades of green. Following its phosphorylation, the R domain dislodges from the nucleotide binding domains (NBDs) dimer interface (step I of the conformational change). This allows the ATP-binding residues to fluctuate and sample conformers near the NBDs-open state (step II), some of which provide an initial ATP-binding site. In step III, this subset of conformers is selected by ATP to initiate binding (i.e. ensemble selection), which then proceeds to complete closure of the NBDs by an induced fit mechanism. In step IV, the channel bursts to the conducting state by a dynamic mechanism, without a major conformational change.

In the absence of phosphorylation and ATP, CFTR adopts solely the NBDs-open conformation and spontaneous conversions to the NBDs-closed conformation are not seen (*Levring et al., 2023*). In addition, multiple phosphorylation events are required to dislodge the R domain from its obstructing position, presenting a very large energetic barrier (*Liu et al., 2017b*; *Gadsby and Nairn, 1999*). This conformational homogeneity of dephosphorylated ATP-free CFTR suggests that the first step of the conformational change, that of movement of the R domain, cannot efficiently proceed via an ensemble selection mechanism. In the absence of definitive experimental/computational evidence, we currently propose that this step occurs by an induced fit mechanism, where the reversible covalent phosphorylation is the allosteric ligand (*Figure 7*, step I). The next step of the conformational change, binding of ATP and closure of the NBDs, is more complex. On the one hand, single molecule studies showed that even after phosphorylation the NBDs solely adopt the open conformation and do not spontaneously close unless ATP is added. Once ATP was added, all the molecules converted to the closed-NBD conformation and were never seen to spontaneously open their NBDs (*Levring et al., 2023*). We observed very similar conformational homogeneity and stability with another member of the ABC transporter family (*Yang et al., 2018*). Along the same lines, MD simulations showed that the closed state with dimerized NBDs is stable over hundreds of nanoseconds (*Zeng et al., 2023*). Taken together, these results strongly argue in favor of an induced fit mechanism, and against MWC/ ensemble selection/population shift models which assume pre-existence of multiple conformations and a spontaneous dynamic equilibrium between them. Direct experimental evidence supporting an induced fit mechanism was also provided by macroscopic current recordings combined with double mutant cycles (*Szollosi et al., 2010*). On the other hand, when the NBDs are separated, the ATP-binding sites are yet to form, it is hard to envision how the nucleotide binds to its non-existing binding site. Such large conformational changes are considered to occur only via an ensemble selection mechanism (*Nussinov et al., 2014*). We suggest that our results provide the solution: In the ATP-free form, despite the separation of the NBDs, the main allosteric sites are located near the ATP-binding sites (*Figures 1 and 2A*). Moreover, many of the dynamic features that manifest following ATP binding exist in a latent form in the ATP-free conformation (*Figure 2B and C*). This means that the ATP-binding residues fluctuate considerably in the absence of nucleotide, sampling conformations in the vicinity of the NBDs-separated native state. Some of these conformations likely provide an initial state for the interaction with the nucleotide (*Figure 7*, step II). ATP will selectively bind to these conformers, as in ensemble selection allosteric mechanism. Once the docking of the nucleotide is initiated by this combination of dynamic fluctuations, ensemble selection closure of the NBDs can proceed by an induced fit mechanism (*Figure 7*, step III) as suggested by experimental evidence (*Szollosi et al., 2010*). The resulting conformational changes do not yet lead to channel opening and conductance: The phosphorylated ATP bound structure observed in the cryo-EM studies is non-conducting: the pore is fully dehydrated, and an ion translocation pathway is missing. Therefore, phosphorylation, ATP binding, and closure of the NBDs are essential yet insufficient for channel opening. Several lines of evidence suggest that the final missing component that enables conductance are the dynamic fluctuations of the NBDs-closed state: MD simulations of chloride conductance showed that a major conformational change is not required for channel opening. Instead, slight side chain re-orientations of pore residues enable sporadic pulses of chloride conductance (*Zeng et al., 2023*). Single molecule measurements also failed to identify any conformational changes between the conducting and non-conducting states (*Levring et al., 2023*). Along the same lines, allosteric potentiators that increase CFTR's channel-open probabilities do so without noticeable effects on its conformation (*Liu et al., 2019*). These studies suggest that the conformations of the conducting and non-conducting states are very similar and that a major conformational change is not required to open the channel. Rather, transition to the conducting state occurs by a dynamic allosteric mechanism with rapid conformational sampling around the phosphorylated ATP-bound NBDs-closed state (*Figure 7*, step IV).

With this combination of allosteric models in mind, we adopted a structural/dynamic view to study the molecular basis of allosteric connectivity in CFTR. The premise of the structural/dynamic perspective is that strain energy generated by binding of the allosteric ligand (phosphorylation and ATP in CFTR) is mechanically transmitted through a network of interacting residues to be relieved by a conformational switch. To identify this network of interacting residues, we used two elastic network approaches: ANM-LD is a trajectory molecular simulation approach which identifies residues that interact during a conformational change (*Acar et al., 2020*). It is therefore particularly useful for

studying induced fit mechanisms that assume the existence of such networks of interacting residues. GNM-TE on the other hand models the intrinsic dynamics of a protein around its mean native conformations and provides a useful tool to study the dynamic/ensemble selection components of CFTR's allostery. Both approaches are computationally economical, and for a protein as large and complex as CFTR provide an attractive alternative to atomistic MD simulations.

Despite their simplifications, and without any input of functional information, the ANM-LD/GNM-TE combination clearly distinguishes the three main functional domains of the protein (i.e. NBDs, gating residues, docking site of the R domain) from the rest of the protein (*Figures 2 and 3*). In further support of the utility of the calculations we observe a remarkable correlation between the residues identified as main allosteric sources and the positions of functionally essential residues (*Figure 1—figure supplement 4*), ATP-binding residues (*Figure 1—figure supplement 5*), and disease-causing mutations (*Figure 1—figure supplement 6*). In addition, despite the high structural similarity between the NBDs, the calculations identify the functional singularity of the catalytic NBD2 and identifies it as the dominant allosteric source (*Figure 2A*). Taken together, these results show that the computations can capture the structural dynamics of CFTR.

The allosteric effects of ATP binding and hydrolysis are not always easily distinguishable. For example, Liu et al. investigated the effects of the force originating from ATP hydrolysis on the chaperonin GroEL. They applied elastic network models and principal component analysis on a series of GroEL structures and found that in this processive enzyme ATP binding is insufficient to induce the conformational changes, which require nucleotide hydrolysis (*Liu et al., 2017a*). In CFTR, a large body of evidence suggests that the allosteric driver is binding per se and not hydrolysis: binding of ATP under non-hydrolyzing conditions is sufficient to dimerize the NBDs and to open the channel (*Zhang et al., 2018*; *Levring et al., 2023*). In addition, mutations and experimental conditions that stabilize the ATP-bound state by reducing hydrolysis rates stabilize the NBDs-closed conformation and conducting state of the channel. At first glance, the allosteric connectivity networks of the ATP-free and -bound conformations seem very different (*Figure 3—figure supplement 1*). However, a closer examination reveals that binding of ATP fine-tunes and brings to the fore dynamics that pre-existed in the ATP-free conformation. For example, in the dephosphorylated ATP-free state conformation, the allosteric signaling originates from many residues in the NBDs, mostly from those that surround the future ATP-binding sites (*Figures 1 and 2*). Binding of ATP focuses these relatively diffuse dynamics to the exact location of the ATP-binding residues (*Figure 3A*). Similarly, in the ATP-free conformation the residues that line the permeation pathway and the gating residues have a minor allosteric role and are detected only once the more dominant fluctuations of the NBDs were omitted from the calculations (*Figure 2B*). Following ATP binding the allosteric role of these residues becomes much more dominant (*Figure 3B*). We propose that in CFTR the role of ATP binding is: (1) to induce the conversion to the NBDs-closed conformation via a combination of ensemble selection and induced fit mechanisms and (2) to modulate conformational dynamics that are present to different degrees in other conformations via a dynamic allosteric mechanism. In CFTR, ATP binding, rather than hydrolysis, is sufficient to exert these effects: binding of ATP is sufficient to induce the closure of the NBDs and to open the channel (*Levring et al., 2023*). ATP hydrolysis and product release are not required for channel opening, but rather lead to separation of the NBDs and termination of conductance. This differs from the role of ATP in processive enzymes such as the chaperonin GroEL, where the exothermic energy released by ATP hydrolysis is required to drive the conformational changes (*Liu et al., 2017a*).

A new allosteric hotspot was identified in TMD1, in a cleft formed by TM helices 2, 3, and 6 (*Figure 3C–D*). Its proximity to the interaction site of the phosphorylated R domain may provide the molecular basis for the dual role of the R domain: an inhibitory role while dephosphorylated, and a conductance-stimulatory role once phosphorylated (*Winter and Welsh, 1997*; *Ostedgaard et al., 2001*; *Csanády et al., 2000*). Perhaps more importantly, this allosteric cleft is highly druggable and provides a new target for structure-guided screening of small molecules.

## Materials and methods

## GNM-based TE (code is freely available at https://github.com/PRC-comp/ANMLD_TE)

GNM is a minimalist 1D elastic network model that assumes a Gaussian distribution of fluctuations that is isotropic in nature, operating within the confines of a given protein's topology. The GNM model relies on a pair of parameters: force constant governing harmonic interactions and the cutoff distance within which these interactions occur. In this model, the force constant is uniform for all interactions and is conventionally set to unity. The range of interaction is defined by a cutoff distance, typically ranging between 7 Å and 10 Å, with a value of 10 Å being employed in this study. TE formulation (*Hacisuleyman and Erman, 2017b*) which is based on GNM (*Haliloglu et al., 1997*; *Bahar et al., 1997*) reveals the direction of information flow between two residues i and j. Given a certain movement in residue i, TE is a measure of the reduction of the uncertainty in the movements of residue j with a time delay $\tau$ between the movements of both residues.

$TE_{i,j}$ ( $\tau$ ) between residues i and j at time $\tau$ is formulated as (*Hacisuleyman and Erman, 2017a*)

$$T_{i \to j}(\tau) = S\left(\Delta R_j(t+\tau) | \Delta R_i(t)\right) - S\left(\Delta R_j(t+\tau) | \Delta R_i(t), \Delta R_j(t)\right) \tag{1}$$

where the s are conditional entropies, given by,

$$S\left(\Delta R_j(t+\tau) | \Delta R_i(t)\right) = -\left\langle \ln p\left(\Delta R_i(0), \Delta R_j(\tau)\right)\right\rangle + \left\langle \ln p\left(\Delta R_i(0)\right)\right\rangle \tag{2}$$

$$S\left(\Delta R_j(t+\tau) | \Delta R_i(t), \Delta R_j(t)\right) = -\left\langle \ln p\left(\Delta R_j(0), \Delta R_j(\tau)\right)\right\rangle + \left\langle \ln p\left(\Delta R_i(0), \Delta R_j(0), \Delta R_j(\tau)\right)\right\rangle \tag{3}$$

For more details of the definitions of GNM time correlation of the residue fluctuations in *Equations 4 and 5*, see *Hacisuleyman and Erman, 2017a*; *Haliloglu et al., 1997*.

The net TE from residue i to j at a given $\tau$ is described as

$$\Delta T_{i \to j}(\tau) = T_{i \to j}(\tau) - T_{j \to i}(\tau) \tag{4}$$

where $\Delta T_{i \to j}(\tau)$ estimates the direction of TE between residues i and j in a certain time delay $\tau$. Thus, the TE calculations reveal entropy/information sources and receivers: Entropy sources have positive net TE values and send information to many other residues, and entropy receivers have negative net TE values and receive information from the rest of the structure.

The choice of $\tau$ between the movement of residues is important to identify causal interrelations: when $\tau$ is too small, only local cause-and-effect relations (between adjacent amino acids) will be revealed. If $\tau$ is big enough, few (if any) long-range cause-and-effect relations will manifest. In a previous work (*Acar et al., 2020*), we studied in detail the effects of choosing different $\tau$ values and found that an optimal value of $\tau$ which maximizes the degree of collectivities of net TE values is in most cases 3× $\tau_{opt}$( $\tau_{opt}$ is the time window in which the total TE of residues is maximized) (see also Figure S1 in *Acar et al., 2020*).

## Degree of collectivity and TECol score

The degree of collectivity is a measure of the number of residues that are influenced by the movement of one residue, i.e., how collective is the information transfer? Utilizing Bruschweiler's study (*Brüschweiler, 1995*), collectivity values of residues in information transfer are calculated based on the positive net TE values (*Altintel et al., 2022*) as

$$K_{i,s} = \frac{1}{N} \exp\left(-\sum_{j=1}^{N} \alpha \left(\Delta T_{ij,s}(\tau)\right)^2 log\left(\alpha \left(\Delta T_{ij,s}(\tau)\right)^2\right)\right) \tag{5}$$

where s is a selected subset of slow GNM modes, N is the total number of residues, and α is the normalization factor which is determined as

$$\sum_{i=1}^{N} a\left(T_{ij,s}(\tau)\right)^2 = 1 \tag{6}$$

Here, s covers 3 subsets of the 10 slowest modes: Modes 1–10 (i.e. all 10), modes 2–10, and modes 3–10. $\tau$ is taken as three times the optimum tau value that maximizes the collective information transfer (**Altintel et al., 2022**). The 10 slowest modes are considered since the eigenvalue distribution of the dynamic modes shows they fully represent the slow end of the dynamic spectrum (**Figure 1— figure supplement 1**).

TECol score of each residue i is calculated by the multiplication of its cumulative positive net TE value (the sum of positive net TE values that residue i sends to the other residues) with its collectivity $K_{i,s}$ value in each subset s as

$$TECol\,Score_{i,s} = \ K_{i,s} \cdot \sum_{j=1}^{N} \Delta T_{ij,s} \tag{7}$$

The TECol score, combining TE and collectivity values, is used in the determination of the most functionally plausible global information source residues that are powerful effectors.

## ANM-LD simulations (code is freely available at https://github.com/PRC-comp/ANMLD_TE)

The conformational change between dephosphorylated ATP-free (PDB ID: 5UAK) (**Liu et al., 2017b**) and ATP-bound (PDB ID: 6MSM) (**Zhang et al., 2018**) human CFTR was simulated using the ANM-LD approach (**Acar et al., 2020**; **Yang et al., 2018**). ANM-LD combines a low-resolution α-carbon-based ANM with stochastic all-atom implicit solvent LD simulations. Intrinsic dynamics characterized by ANM dynamic modes and followed short-time LD simulations guide the conformational transition from the initial to the target state. That is, the conformational transition pathway is successfully evolved from the dephosphorylated ATP-free to the phosphorylated ATP-bound CFTR, starting with 7.5 Å and converging to 2.3 Å of RMSD. Due to the lack of a characterized structure, the regulatory domain is not included in the ANM-LD simulations. The time window in ANM-LD simulations does not reflect real time of simulated transitions yet provides information on transition trajectory and sequence of events, using accessible internal dynamics as the only bias toward the target conformation.

### ANM-LD simulation flow

The hessian matrix (**H**) with a distance threshold radius ($R_{cut}$ = 13 Å) is constructed at each cycle of the ANM-LD simulation to calculate 3N-6 ANM dynamic modes, i.e., eigenvectors $U_k$ and eigenvalue $\lambda_k$, k=1, 3N-6, where N is the number of residues (**Atilgan et al., 2001**). The ANM mode that best overlaps with the difference vector of the aligned initial/intermediate and target conformations is selected ($U_{best}$) and the initial conformation is deformed along this vector using a deformation factor (DF: 0.4 Å). Energy minimization is performed for 500 steps and 100 steps of LD simulations are performed with a time step of 0.2 fs at a temperature of 310 K using the Amber 11-Sander 4 biomolecular simulation program (**Case and Aktulga, 2010**).

The simplified position conformational sampling of ANM-LD cycles is given below as

$$\boldsymbol{R}_{new} = \boldsymbol{R}_{old} + \Delta\boldsymbol{R}_{ANM} + \Delta R_{Eng.Min} + \Delta\boldsymbol{R}_{LD} \tag{8}$$

where $\Delta\boldsymbol{R}_{ANM} = \boldsymbol{U_{best}} \cdot DF$.

$\Delta\boldsymbol{R}_{Eng.Min}$ and $\Delta\boldsymbol{R}_{LD}$ are energy minimization and LD simulations contributions to the sampling, respectively. The ANM steps and LD steps are iteratively performed for a predetermined number of ANM-LD cycles or until the RMSD between intermediate and target states converges. In the present ANM-LD simulations of CFTR the RMSD converged to a value of 2.3Å.

### Equal time and time delay cross-correlations

Each residue has a fluctuation vector $\Delta\boldsymbol{R}$ that corresponds to the difference between its current position to the mean position of its α-carbons. The mean positions are calculated from the selected time window of the ANM-LD simulation cycles. The pseudo correlation between the fluctuations of residues i and j is calculated as

$$C_{ij}(\tau) = \frac{< \Delta \boldsymbol{R}_i(t) \cdot \Delta \boldsymbol{R}_j(t+\tau) >}{\sqrt{< \Delta \boldsymbol{R}_i(t)^2 >} \sqrt{< \Delta \boldsymbol{R}_j(t+\tau)^2 >}} \tag{9}$$

where $t$ represents an instantaneous pseudo time and $\tau$ is an additive time delay in ANM-LD cycles. When $\tau$ equals to 0, $Cij$ reduces to standard correlation between the movements of residues $i$ and $j$, with values between –1 and 1, where 0 corresponding to no correlation, and positive and negative values corresponding to movement in the same or opposite directions, respectively. When $\tau$ is greater than 0, the correlation between the fluctuations of residues $i$ at time $t$ and $j$ at $t+\tau$ indicates that the movement of residue $i$ is leading and of residue $j$ is following in the case of $C_{ij}(\tau) > C_{ji}(\tau)$. The time window in the conformational transition (part of the ANM-LD simulation trajectory from which the conformations are extracted) and the time delay $\tau$ are the two main parameters in defining the correlation behavior.

The time delay, $\tau$, an order parameter in a running time window, is a measure of changes in the dynamics that allows a temporal coarse graining in the transition pathway, i.e., quantifying directional correlations. Leading/following events, depending on the length of the time delay and the time window, reveal cues about the complex hierarchical reorganization of residues in their motion. Here, a time delay of 16 cycles is taken in a transition window of 48 cycles from the ATP-free to the ATP-bound states of CFTR. This time delay corresponds to a time window at which the autocorrelation average over all residues shifts from a positive value to a negative value with respect to the mean conformation within 48 cycles.

## Statistical analysis

In dephosphorylated ATP-free human CFTR (PDB ID 5UAK), we identified a total of 60 allosteric peaks with (1) above average TECol score and (2) above average peak prominence. We then counted how many times the positions of the functionally essential residues or ATP-binding sites spatially correlate with these peaks using cutoff distances of 0 Å (i.e. exact match), 4 Å, or 7 Å. To evaluate the statistical significance of these predictions, we generated 100,000 sets of 60 randomly localized allosteric peaks and counted how many times the positions of the functionally essential residues or ATP sites were in the vicinity of these random allocations. Thus, $Z$ scores of the predictions were determined using the definition of

$$Z = \frac{X - \mu}{\sigma} \tag{10}$$

where X is the count of correct guesses of our prediction while μ and σ are the mean and the standard deviation of the correct matches of random samples, respectively. From the obtained $Z$ scores, p-values of the predictions were calculated by one-tailed hypothesis test using a significance level of 0.05.

The same statistical analysis is performed for the spatial correlation of the hinge positions with the entropy source peaks using cutoff distances of 4 Å or 7 Å.

## Visualization

The 3D structural figures presented in this work are created using PyMol 2.5.4 (The PyMOL Molecular Graphics System).

## Acknowledgements

This work was supported by grants from NATO Science for Peace and Security Program (SPS project G5685, to OL, TH, and NB-T), the Israeli Academy of Sciences project 1006/18 (OL), and The Scientific and Technological Research Council of Turkey (TUBITAK) grant number 119F392 (TH).

# Additional information

## Funding

| Funder | Grant reference number | Author |
|---|---|---|
| North Atlantic Treaty Organization | Science for Peace G4622 | Ayca Ersoy |
| Israel Academy of Sciences and Humanities | 1006/18 | Nurit Livnat Levanon |
| Scientific and Technological Research Council of Turkey | 119F392 | Turkan Haliloglu |

The funders had no role in study design, data collection and interpretation, or the decision to submit the work for publication.

## Author contributions

Ayca Ersoy, Bengi Altintel, Data curation, Formal analysis, Validation, Investigation, Visualization, Methodology; Nurit Livnat Levanon, Data curation, Formal analysis, Validation, Investigation, Visualization; Nir Ben-Tal, Formal analysis, Supervision, Funding acquisition, Project administration, Writing – review and editing; Turkan Haliloglu, Data curation, Formal analysis, Supervision, Funding acquisition, Investigation, Methodology, Writing – original draft, Project administration, Writing – review and editing; Oded Lewinson, Conceptualization, Formal analysis, Supervision, Funding acquisition, Validation, Investigation, Visualization, Methodology, Writing – original draft, Project administration, Writing – review and editing

## Author ORCIDs

Nir Ben-Tal (ID) https://orcid.org/0000-0001-6901-832X
Oded Lewinson (ID) https://orcid.org/0000-0003-4650-0441

Reviewer #1 (Public Review): https://doi.org/10.7554/eLife.88659.3.sa1
Reviewer #2 (Public Review): https://doi.org/10.7554/eLife.88659.3.sa2
Author Response https://doi.org/10.7554/eLife.88659.3.sa3

---

# Additional files

## Supplementary files
• MDAR checklist

## Data availability

The current manuscript is a computational study. The code for GNM-TE is freely available on GitHub (copy archived at *PRC-comp, 2023*).

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
