## [Editor Report · eLife assessment]

This manuscript presents a **useful** analysis of allosteric communication in the CFTR protein using a coarse-grained dynamic model and characterized the role of disease-causing mutations. The results and analyses are generally **solid** and validated with available experimental observations. The findings provide comprehensive insights into the allosteric mechanism of this protein.

---

## [Referee Report · Reviewer #1 (Public Review)]

The paper offers interesting insight into the allosteric communication pathways of the CTFR protein. A mutation to this protein can cause cystic fibrosis and both synthetic and endogenous ligands exert allosteric control of the function of this pivotal enzyme. The current study utilizes Gaussian Network Models (GNMs) of various substrate and mutational states of CFTR to quantify and characterize the role of individual residues in contributing to two main quantities that the authors deem important for allostery: transfer entropy (TE) and cross correlation. I found the TE of the Apo system and the corresponding statistical analysis particularly compelling. The authors updated the manuscript nicely to include the limitations of the chosen model (GNM) and thus allow the reader to assess the limitations of the results. I appreciated the comprehensive discussion of a proposed mechanism by which allostery is achieved in the protein (though I would have put that in the introduction and had it motivate the choice of methods). This discussion allows the reader to place the allosteric mechanism of this protein in the broader context of protein allostery.

---

## [Referee Report · Reviewer #2 (Public Review)]

In this study, the authors used ANM-LD and GNM-based Transfer Entropy to investigate the allosteric communications network of CFTR. The modeling results are validated with experimental observations. Key residues were identified as pivotal allosteric sources and transducers and may account for disease mutations.

The paper is well written and the results are significant for understanding CFTR biology.

---

## [Author Response]

The following is the authors’ response to the original reviews.

**Reviewer #1 (Public Review):**
The paper offers some potentially interesting insight into the allosteric communication pathways of the CTFR protein. A mutation to this protein can cause cystic fibrosis and both synthetic and endogenous ligands exert allosteric control of the function of this pivotal enzyme. The current study utilizes Gaussian Network Models (GNMs) of various substrate and mutational states of CFTR to quantify and characterize the role of individual residues in contributing to two main quantities that the authors deem important for allostery: transfer entropy (TE) and cross correlation. I found the TE of the Apo system and the corresponding statistical analysis particularly compelling. I found it difficult, however, to assess the limitations of the chosen model (GNM) and thus the degree of confidence I should have in the results. This mainly stems from a lack of a proposed mechanism by which allostery is achieved in the protein. Proposing a mechanism and presenting logical alternatives in the introduction would greatly benefit this manuscript. It would also allow the authors to place the allosteric mechanism of this protein in the broader context of protein allostery.

As detailed below, we went to great lengths to address these concerns, with an emphasis on the limitations of the model and a proposed mechanism. These revisions should hopefully warrant are-evaluation of our manuscript.

**Reviewer #1 (Recommendations For The Authors):**
1. It would greatly benefit the paper to state a proposed mechanism by which allostery is achieved in this protein. Is this through ensemble selection, ensemble induction, or a purely dynamic mechanism? What is the rationale for choosing the proposed mechanism and what are reasonable alternative mechanisms? How does this mechanism fit in the broader context of protein allostery?

Following this comment, we added a VERY extensive description of the proposed mechanism by which allostery is achieved in CFTR and present the rationale for choosing this mechanism (lines 445-97 and Figure 7). Briefly, based on previous experimental results and our results we propose that no single model can explain allostery in CFTR, and that its allosteric mechanism is a combination of induced fit, ensemble selection, and a dynamic mechanism.

2. With a proposed mechanism in place, the choice of a GNM to investigate the mechanism and eliminate alternative mechanisms should be rationalized.

The rationale for choosing GNM (and ANM-LD) to study the proposed mechanism is now given in lines 498-510. Please note however, that as mentioned in the response to point 1 (and detailed in lines 445-97), the choice of allosteric mechanism, and ruling out other alternatives was not based solely on GNM and ANM-LD, but also on previous experimental results.

3. A discussion of the strengths and limitations of the GNM are pivotal to understanding the limitations of the results shown. How sensitive are the results to specific details of the model(s)?

a. A discussion of the strengths and limitations of the GNM have been added to the introduction. Please see lines 107-122.

b. Sensitivity of the results to the specific details of GNM:

GNM uses two parameters: the force constant of harmonic interactions and the cutoff distance within which the existence of the interactions is considered. The force constant is uniform for all interactions and is taken as unity. Its value affects only the absolute values of the fluctuations (i.e., their scale) but not their distribution. As we are only looking at fluctuations in relative terms our results are insensitive to its value. GNM uses a cutoff distance of 7-10 Å in which interactions are considered (10 Å used in this study). To test the sensitivity of the results to the cutoff distance we repeated the calculations using 7 Å. As now discussed in lines 170-73 and shown in Figure S2 the results remained largely unchanged.

c. Sensitivity of the results to the specific details of TE: To identify cause-and-effect relations TE introduces a time delay (τ) between the movement of residues. The choice of τ is important: when τ is too small, only local cause-and-effect relations (between adjacent amino acids) will be revealed. if τ is too big, few (if any) cause-and-effect relations will manifest. This is analogous to the effects of a stone throne into a lake: look too soon, before the stone hits the water, and you’ll see no ripples. Look too late, the ripples will have already subsided. In a previous work (PMID 32320672), we studied in detail the effects of choosing different τ values and found that an optimal value of τ which maximizes the degree of collectivities of net TE values is in most cases 3× τopt (τopt is the time window in which the total TE of residues is maximized). Details of how τ was chosen were added to the methods section.

In general, the limitations of the chosen model(s) is difficult to determine from the current manuscript because it is devoid of details of the model. While I understand that GNMs have been widely used to study protein systems, the specifics of the model are central to the current work and thus should be provided somewhere in the manuscript.

a. As mentioned in our response above, the limitations of GNM are now presented (lines 107-122).

b. The specifics of the model are now given in more detail in the methods section.

c. In addition, as mentioned above, the results are largely independent of the values of the model’s parameters.

b. Would changing the force constants to a more anisotropic model qualitatively change the results?

a. GNM assumes isotropic fluctuations, and the calculations are based on this assumption. Therefore, GNM is inherently an isotropic model.

b. Importantly, we complement the GNM-TE calculations with ANM-LD simulations, which predict the normal modes in 3D using an anisotropic network model.

4. How repeatable is the difference between no ATP bound and ATP bound CFTR? I worry that the differences in TE in Figures 1 and 3A are mainly due to two different crystallization conditions. Is there evidence that two different structures of the same protein in the same ligand state lead to small changes in TE?

To address this concern, we repeated the calculations using the structures of the ATP-free and bound forms of zebrafish CFTR. As now explained in text (lines 298-303) and shown in Figure S8 the effects of ATP are highly repeatable.

5. Collective modes - why should we expect allostery to be in the most collective modes? Let alone the 10 most? Why not do a mode by mode analysis? Why, for example, were two modes removed page 9 first full paragraph?

a. Collective modes: We have erroneously referred to the slow modes as collective modes.This has now been corrected throughout the manuscript.

b. Let alone the 10 most?

c. why should we expect allostery to be in the most collective modes? Residues that are allosterically coupled are expected to display correlated motions. The slow modes (formerly referred to as “collective modes”) are generally the most collective ones, i.e., display the greatest degree of concerted motions. We therefore expect these modes to contain the allosteric information.

d. Furthermore, as now explained in the text (lines 163-69) and in Figure S1 the Eigenvalue decays of ATP-free and -bound CFTR demonstrate that the 10 slowest GNM modes sufficiently represent the entire dynamic spectrum (the distribution converges after the 10th slow mode).

e. Why not do a mode by mode analysis? It is entirely possible to do a mode-by-mode analysis. However, our view is that the allosteric dynamics of a protein is best represented by an ensemble of modes, rather than by individual ones. We found (as detailed here PMID 32320672) that it is more informative to first use the complete set of modes that encompasses the dynamics (the 10 slowest modes in our case) and then gradually remove the dominant modes.

f. As explained in text (lines 254-7) and more elaborately in our previous work (PMID35644497), the large amplitude of the slowest modes may hide the presence of “faster” modes that may nevertheless be of functional importance. Removal of the 1-2 slowest modes often helps reveal such modes.

g. Why, for example, were two modes removed page 9 first full paragraph? As explained for the ATP-free form (lines 257-60), removal of these two slowest modes allowed the “surfacing” of dynamic features which were hidden before. We propose that these dynamic features are functionally relevant (see lines 304-19). Removal of other modes did not provide additional insight.

Minor issues:1. Statements like "see shortly below" should be made more specific (or removed completely).

Corrected as suggested

2. "interfered" should be "inferred" page 10 middle of the first full paragraph

Corrected as suggested

3. End parenthesis after "for an excellent explanation about the correlation between TE and allostery see (41)." Page 4 middle of first full paragraph

Corrected as suggested

**Reviewer #2 (Public Review):**
In this study, the authors used ANM-LD and GNM-based Transfer Entropy to investigate the allosteric communications network of CFTR. The modeling results are validated with experimental observations. Key residues were identified as pivotal allosteric sources and transducers and may account for disease mutations.The paper is well written and the results are significant for understanding CFTR biology.
**Reviewer #2 (Recommendations For The Authors):**
Technical comments:p4 Please explain how is the time delay parameter tau chosen (ie. three times the optimum tau value...)? It seems this unknown time should depend on the separation between i and j. Is the TE result sensitive to the choice of tau? How does the choice of cutoff distance of GNM affect the TE result?

a. The choice of τ is important: when τ is too small, only local cause-and-effect relations (between adjacent amino acids) will be revealed. if τ is too big, few (if any) cause-and-effect relations will manifest. This is analogous to the effects of a stone throne into a lake: look too soon, before the stone hits the water, and you’ll see no ripples. Look too late, the ripples will have already subsided. In a previous work (PMID 32320672), we studied in detail the effects of choosing different τ values and found that an optimal value of τ which maximizes the degree of collectivities of net TE values is in most cases 3× τopt (τopt is the time window in which the total TE of residues is maximized). Details of how τ was chosen were added to the methods section.

b. To test the sensitivity of the results to the cutoff distance we repeated the calculations using 7 Å. As now discussed in lines 170-173 and shown in Figure S2 the results remained largely unchanged.

It would be nice to directly validate the causal prediction by GNM-based TE. For example, is it in agreement with direct causal observation of MD simulation? If the dimer is too big for MD, perhaps MD is more feasible for the monomer (NBD1+TMD1).

a. The causality we determined using GNM-based TE is in good agreement with conclusions drawn from single channel electrophysiological recordings and rate-equilibrium free-energy relationship analysis (Sorum et al; Cell 2015, and see lines 8691, and 364-70).

b. To the best of our knowledge, causality relations in CFTR are yet to be determined by MD simulations (This is likely because the protein is too big and the conformational changes are very slow). We cannot therefore compare the causality.

c. Conducting MD simulations on half of CFTR (NBD1+TMD1) is not likely to be very informative: the ATP binding sites are formed at the interface of NBD1 and NBD2, and the ion translocation pathway at the interface of the TMDs.

p5 How are the TE peak positions different from other key positions as predicted by GNM, suchas the hinge positions with minimal mobility of the dominant GNM modes?

Following this comment, we compared the positions of the GNM-TE peaks and the hinge positions as determined by GNM. As now discussed in lines 173-178 and shown in Figure S3 we observed partial overlap which was nevertheless statistically significant (Figure S3).

p7 How to select the 10 most collective GNM modes? Why not use the 10 slowest GNM modes?

We have actually used the 10 slowest GNM modes, but in an attempt to cater for the non-specialist reader, we referred to them as the most collective ones. This has now been corrected throughout the manuscript and the terminology that is now used is “10 slowest modes”

p9 There exist other ANM-based methods for conformational transition modeling. So it would be nice to discuss their similarity and differences from ANM-LD, and compare their predictions.

Alternative ANM (and other elastic network models) -based methods are now mentioned and referenced in lines 144-50. These methods are different from ANM-LD in the details of the all atom simulations and in their integration with the elastic network model. It is not trivial to reanalyze CFTR’s allostery using these methods and is beyond the scope of this work.

Regarding the prediction of order of residue motions, can one directly observe such order by superimposing some intermediate conformation of ANM-LD with the initial and end structure?

This would indeed be very attractive approach to visualize the order of events and following this comment we have tried to do just so. Unfortunately, we failed: Superimposing pairs of frames provided little insight, and we therefore compiled a video comprising all frames, or videos based on averages of several time delayed frames. We found that it is next to impossible to discern (using the naked eye) the directionality of the fluctuations and follow the order of conformational changes. Therefore, at this point, we have abandoned this endeavor.

**Reviewer #3 (Public Review):**
This study of CFTR, its mutants, dynamics, and effects of ATP binding, and drug binding is well written and highly informative. They have employed coarse-grained dynamics that help to interpret the dynamics in useful and highly informative ways. Overall the paper is highly informative and a pleasure to read.The investigation of the effects of drugs is particularly interesting, but perhaps not fully formed.This is a remarkably thorough computational investigation of the mechanics of CFTR, its mutants, and ATP binding and drug binding. It applies some novel appropriate methods to learn much about structure's allostery and the effects of drug bindings. It is, overall, an interesting and well written paper.There are only two main questions I would like to ask about this quite thorough study.
**Reviewer #3 (Recommendations For The Authors):**
1. Is it possible that the relatively large exothermic ATP hydrolysis itself exerts a force that causes the observed transitions? Jernigan and others have explored this effect for GroEL and some other structures. The effects of ATP binding and hydrolysis are likely often confused, and both are likely to be important.

It is well established by many studies that ATP hydrolysis is not required to drive the conformational changes or to open the channel, and that ATP binding per-se is sufficient e.g., We have clarified this point in lines 521-30.

2. For the case of ivacaftor, would a comparison of the motion's directions show that ivacaftor might be compensating simply by its mass being located in a site to compensate for the mass changes from the mutations (ENMs with masses needed to address this). We have observed such cases on opposite sides of a hinge.

We do not think that this is the case, from the following reasons:

a. Ivacaftor corrects many gating mutations (e.g., G551D, G178R, S549N, S549R, G551S, G970R, G1244E, S1251N, S1255P, G1349D) which are spread all over the protein. Ivacaftor binds to a single site in CFTR, and it is therefore unlikely that its mass contribution corrects all these diverse mass changes.

b. The residues that comprise the Ivacaftor binding were identified as allosteric “hotspots” in both the ATP-free and -bound forms (Figures 2B, 3B, and 6A), also in the absence of the drug. This indicates that the dynamic traits of this site is intrinsic to it, and that once bound, the drug acts by modulating these dynamics

The Abstract does not repeat some of the more interesting points made in the paper and would benefit from a substantial revision.

Corrected as suggested

There are just a few minor points (just words):P 3 line 2 of first full ¶: "effects" should be "affects"

Corrected as suggested

P 6 first line "per-se" should be "per se"

Corrected as suggested

Further down that page "two set" should be "two sets"

Corrected as suggested

Even further down that same page "testimony" should be "support"

Corrected as suggested

P 10, 5 lines from the bottom "impose that" is awkward

Changed to “define”